# SurroCBM: Concept Bottleneck Surrogate Models for Label-free Post-hoc Explanation

## Abstract

Explainable AI seeks to bring light to the decision-making processes of black-box models. Traditional saliency-based methods, while highlighting influential data segments, often lack semantic understanding. Recent advancements, such as Concept Activation Vectors (CAVs) and Concept Bottleneck Models (CBMs), offer concept-based explanations but necessitate human-defined concepts. However, human-annotated concepts are expensive to attain. This paper introduces the Concept Bottleneck Surrogate Models (SurroCBM), a novel framework that aims to explain the black-box models with automatically discovered concepts. SurroCBM identifies shared and unique concepts across various black-box models and employs an explainable surrogate model for post-hoc explanations. An effective training strategy using self-generated data is proposed to enhance explanation quality continuously. Through extensive experiments, we demonstrate the efficacy of SurroCBM in concept discovery and explanation, underscoring its potential in advancing the field of explainable AI.

## 1 Introduction

Explainable AI aims to explain the decision-making process of black-box models. A traditional approach to achieving this transparency is through the use of saliency-based methods which identify the most influential segments of the input data that contribute significantly to a model's decision. Although saliency-based methods highlight the important regions, they do not necessarily offer a semantic understanding. A recent stream of methods, concept-based explanation, aims to use a set of concepts with high-level human-understandable meanings to explain model decisions. Kim et al. (2018) in-

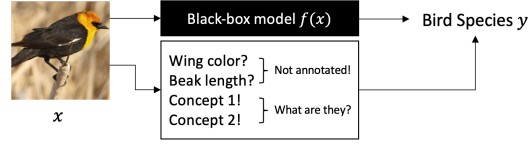

Figure 1: An example of the problem in this paper. The black-box classifier's decisions can be explained with a set of concepts, but they require human labor to annotate and are often hard to attain. We aim to explain the black-box model's behavior with a set of concepts discovered by ourselves.

troduced the Concept Activation Vectors (CAVs), vectors in the activation layer in the direction of user-given concepts, and quantify their importance to the predictions to explain model decisions. Koh et al. (2020) designed a type of self-explainable neural network, Concept Bottleneck Models (CBMs), which first use the data to predict concept values, then predict the targets with concepts, to make the decision-making process more transparent. However, both these two types of concept-based explanation methods require human-defined concepts, which are costly to attain. Some research focuses on post-hoc explanations with incomplete concepts. Yuksekgonul et al. (2022) proposed a method to transfer annotated concepts from other datasets or leverage multimodal models to attain concept annotations for post-hoc explanation. Moayeri et al. (2023) proposed to extract concept activation vectors from text with CLIP model and use them for model explanations. However, since they adopted the idea of borrowing concepts from other data, these works did not thoroughly solve the problem of human labor for concept annotations.

In this work, our goal is to explain the black-box model decisions with automatically discovered concepts, as shown in Fig. 1. Although some results on concept-based model explanation and concept discovery has been encouraging, this task is still challenging due to the following reasons:

**Challenge 1: Bridging the Gap Between Concepts for Data and Post-hoc Explanations.** There is an inherent gap between the explainable concepts to underly the dataset and the related concepts to explain the decision-making process of the black-box models. Existing work typically pays efforts to one of two goals: (1) discovering concepts to explain a dataset, requiring the concepts to be human-understandable, disentangled and fully cover each varying aspect of the dataset; (2) Using given concepts to explain the decision-making processes of a classifier, requiring the concepts to have information for the classification task. With different goals, the required concepts should have different meanings. The different requirements to explain the data and classifiers bring a challenge in discovering concepts that meet the post-hoc explanation requirements.

**Challenge 2: Aligning the Shared Related Concepts for Multiple Classifiers.** While the majority of research focuses on identifying concepts to only explain the data Kim & Mnih (2018) or explain a singular classifier O'Shaughnessy et al. (2020); Tran et al. (2022), real-world applications often require predicting several aspects of the same data. The groups of concepts related to different tasks are different. It is challenging to identify shared and unique concepts that underpin the decision-making processes of multiple classifiers, especially during the concept discovery process.

**Challenge 3: Ensuring High Fidelity of Surrogate Models.** Surrogate model-based explanation methods require high fidelity to mimic the black-box models to ensure accuracy. However, with a limited training set, it is hard to fully mimic the output of the black-box models with surrogate models. Moreover, the input of the surrogate model, defined by the discovered concepts, may not cover all aspects of the original input data, making it more difficult to maintain fidelity.

To tackle these challenges, we introduce the Concept Bottleneck Surrogate Models (SurroCBM), a surrogate model-based method to jointly solve the unsupervised concept discovery and post-hoc explanation problem. Our model can discover the shared and unique concepts across different black-box models on the same data. Our concept-based explainer first maps the data to concepts then identifies the task-related concepts and predicts the black-box model output with a highly transparent module. The contributions of this paper are summarized as follows:

- **A novel framework for discovering identifiable and task-related concepts.** Our proposed method discovers identifiable concepts with relations to multiple classifiers by aligning the shared concepts and identifying the unique concepts of each prediction target.

- **A concept-based post-hoc explainer for black-box model explanation.** Our proposed surrogate model first maps the data to concepts, then identifies a group of related concepts and uses them to explain the model behaviors, providing a high explainability.

- **A training strategy to increase the fidelity with generated data.** In order to continuously enhance the fidelity of the surrogate model, we propose a training strategy that generates user-customizable and diversified additional data to train the model.

## 2 RELATED WORK

### 2.1 CONCEPT DISCOVERY

The unsupervised concept discovery problem aims to identify concepts without given concept labels. Tradition works focus on the form of concepts as important vectors in the activation space Kim & Mnih (2018). Later concept discovery methods aim to identify meaningful image segmentations as concepts Ghorbani et al. (2019); Wang et al. (2023); Yao et al. (2022); Kamakshi et al. (2021); Posada-Moreno et al. (2022), and use them to explain the model behaviors. Another type of method is to use latent factors of generative models as concepts and conduct interventions to get their semantical meanings O'Shaughnessy et al. (2020); Tran et al. (2022). Some more recent work focuses on identifying text descriptions to explain the data and model decisions Yang et al. (2023); Oikarinen et al. (2023); Moayeri et al. (2023).

### 2.2 CONCEPT-BASED EXPLANATION

Our method can be categorized as post-hoc explainability for deep learning models based on concepts. The term *concepts*, there are various definitions, such as a direction in the activation space, a prototypical activation vector, or a latent factor of a generative model. For example, a generative

model such as VAEs Kingma & Welling (2013) can provide a concept-based explanation as it learns a latent representation that captures different aspects of the data. However, standard VAEs struggle to disentangle latent concepts due to their lack of explicit mechanisms for separating intertwined factors of variation, leading to overlapping or mixed representations in the latent space. Concept Activation Vectors (CAVs) Kim et al. (2018) provide an interpretation of a neural net's internal state in terms of human-friendly concepts by viewing the high-dimensional internal state of a neural net as an aid, not an obstacle. ConceptSHAP Yeh et al. (2020) infers a complete set of concepts that are additionally encouraged to be interpretable by retraining the classifier with a prototypical concept layer. O'Shaughnessy et al. (2020) generates causal post-hoc explanations of black-box classifiers based on a learned low-dimensional representation of the data.

## 3  PROBLEM FORMULATION

In this work, we aim to explain the black-box classifiers with automatically identified concepts. We aim to identify a set of concepts that have the ability to act as units of high-level features of data and high-level reasoning for classifications on the data, and discover how the learned concepts combine to explain black-box classifiers for the data.

More formally, given a dataset $\mathcal{X}$ and a set of black-box classifiers $f = \{f_1, f_2, ..., f_{k_y}\}$, each mapping from $\mathcal{X}$ to a target $\mathcal{Y}_i \in \mathcal{Y}$. Our goal is to (1) identify a set of concepts with the values $z = \{z_1, z_2, ..., z_{k_c}\} \in \mathcal{Z} \subset \mathbb{R}^{k_c}$, where $k_c$ denotes the number of concepts, that can serve as reasoning units of $f$; and (2) find a mapping $h : \mathcal{Z} \to \mathcal{Y}$ that map the concept values to the black-box model outputs with a more explainable inner structure, thus it can mimic the black-box model behaviors and provide post-hoc explanations.

To achieve this goal, several challenges of the discovered concepts and the post-hoc explanation process are identified as follows:

- *Fidelity*: To make the post-hoc explanations reliable, predictions derived from concepts via the mapping $h$ must closely mimic the behavior of the black-box models.
- *Identifiability*: To allow the identified concepts to explain new classifiers unseen in training, the identified concepts should comprehensively cover the aspects of data. This requires that the data can be recovered with its corresponding concept values.
- *Explainability*: To provide human-understandable explanations, the explanation process of predicting the targets from identified concepts should be transparent and explainable.

## 4  CONCEPT BOTTLENECK SURROGATE MODELS

**Overview.**  We devised a novel method, Concept Bottleneck Surrogate Models (SurroCBM), to jointly discover the high-level concepts with our desired properties and use the discovered concepts to explain the black-box models. The proposed framework is illustrated in Fig. 2. We first present the model architecture and how the model can be used for local and global explanations in Sec. 4.1, then we induce the training objective in Sec. 4.2 and present a procedure to continuously increase the fidelity in Sec. 4.3.

### 4.1  PROPOSED MODEL

Specifically, we use a surrogate model $f'$ to mimic the behaviors of the black-box model $f$, where $f$ and $f'$ are both a set of classifiers. Inspired by traditional Concept Bottleneck Models, we divide $f'$ into two stages: a concept extractor $e_\phi$ to map the data to concept values $z$, and an explainable mapping $h$ to map the concept values $z$ to the model output. To ensure identifiability, we add an additional decoder $g_\theta$ to map the concept values $z$ back to the data $x$ and minimize their difference. The surrogate model is shown in Fig. 2 (a).

To further improve the explainability of the surrogate model $f'$, we design an explainable interior structure for the mapping $h$, which can identify shared and unique concepts required for each classification target, shown in Fig. 2 (b). This mechanism is implemented with a trainable binary mask $m \in \{0, 1\}^{k_z \times k_y}$, named *explanation mask*. After the model is well trained, we expect the masked

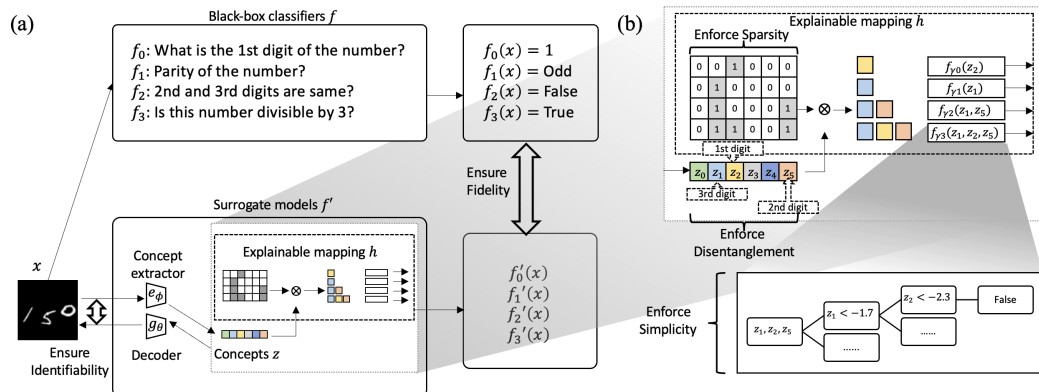

Figure 2: The illustration of our proposed framework. We use the surrogate models $f'$ to mimic the black-box models $f$'s behaviors for post-hoc explanation. In the surrogate model, the data $x$ is first mapped to its concept values $z$ with concept extractor $e_\phi$, then the concept values $z$ are used to predict the model output with an explainable mapping $h$. The mapping $h$ achieves a high explanability by identifying the related concepts to each target and using a soft decision tree to enhance the transparency.

concepts will keep only the ones with relationships to each classification target. The whole set of concept values $z$ is first masked with $m$ with an element-wise product. Thus the input of the estimators $f_\gamma$ will only contain a set of necessary concepts specific to each target. We use soft decision trees to implement each $f_\gamma$ to enhance the explainability of the mapping from related concepts to model outputs.

After the model is trained, the semantic meanings of these concepts are derived through interventions within the generative process, serving as base units for explaining the decision-making process. Below, we discuss the procedure for both global and local explanations.

**Global explanation.** Our method can provide global explanations by identifying the related concepts for each prediction task. For a specific task $f_k$, where $k$ is the index of the task, the related concepts can be identified by

$$z_k^R = \{z_j\}_{m_{j,k}=1} \tag{1}$$

where $z_k^R$ represents the related concepts of the task with the index $k$, and $z_j$ denotes the concept variables (without specified values).

**Local explanation.** Our proposed method can also provide a local explanation of the decision-making process of each data sample's classification. This is achieved by first identifying the related concepts using the global explanation method, and then extracting the values of the concepts with the concept extractor. By feeding the combinations of concepts into the decision tree, which maps concepts to predictions, we ensure transparency in the rules of every decision-making step and its associated predictions for specific data.

Formally, our proposed surrogate model can bring light on the decision-making process of the data sample $x$ on the black-box model $f_k$ with 1) related concepts $z_k^R = \{z_j\}_{m_{j,k}=1}$, 2) values of related concepts: $z_k^R = \{e_\phi(x)_j\}_{m_{j,k}=1}$ and 3) a decision tree from related concepts to predictions: $y = f_{\gamma k}(z_k^R)$.

## 4.2 TRAINING OBJECTIVE

In order to optimize our proposed model, three criteria should be satisfied: (1) the decoded data from concepts should be accurately recovered to match the original data, (2) the predictions from the surrogate model should closely align with the predictions from the black-box model, and (3) the mapping from concepts to predicted labels should be explainable. We derive three corresponding loss terms for them, namely *identifiability loss* ($\mathcal{L}_I$), *fidelity loss* ($\mathcal{L}_F$) and *explainability loss* ($\mathcal{L}_E$).

Then the objective can be written as

$$\min_{\phi,\theta,\gamma,m} \underbrace{\mathcal{L}_I(x;\phi,\theta)}_{\text{Identifiability loss}} + \lambda_1 \underbrace{\mathcal{L}_F(x,f;\phi,m,\gamma)}_{\text{Fidelity loss}} + \lambda_2 \underbrace{\mathcal{L}_E(x;\phi,m,\gamma)}_{\text{Explainablity loss}} \qquad (2)$$

where $\mathcal{L}_I$, $\mathcal{L}_E$, and $\mathcal{L}_F$ denote the corresponding terms for Identifiability loss, Explainability loss, and Fidelity loss. $\phi,\theta,\gamma,m$ are the weights of the corresponding model components.

### 4.2.1 FIDELITY AND IDENTIFIABILITY

Ensuring fidelity requires the output of the surrogate model should be close to the output of the black-box model. So we can naturally use

$$\mathcal{L}_F(x,f;\phi,m,\gamma) = \mathcal{D}(h_{m,\gamma}(e_\phi(x)), f(x)) \qquad (3)$$

where $\mathcal{D}$ is a measure of distance between the black-box model output $f(x)$ and the surrogate model output $h_{m,\gamma}(e_\phi(x))$.

To ensure identifiability, we propose to model the generative process of how the concept values $z$ can be mapped back to the original data $x$. We denote the generative process as $p_\theta(x|z)$. To infer $z$, we use $q_\phi(z|x)$ with learnable weights $\phi$ to estimate the posterior distribution of $p(z|x)$. To ensure the data $x$ can be recovered given concept values $z$, we maximize the variational lower bound on the log-likelihood $p_{\theta,\phi}(x)$. Given the approximated posterior $q_\phi(z|x)$, which naturally matches the objective of Variational Autoencoders. Thus the identifiability loss can be written as:

$$\mathcal{L}_I(x;\phi,\theta) = \mathbb{E}_{q_\phi(z|x)} \log p_\theta(x|z) - D_{KL}(q_\phi(z|x)\|p(z)) \qquad (4)$$

where $p(z)$ is the prior distribution of $z$, and $D_{KL}$ stands for the Kullback–Leibler divergence.

### 4.2.2 EXPLANABILITY

In order to improve the explainability of the surrogate model, we aim to enforce (1) the disentanglement of concepts, and (2) the explainability of the mapping from extracted concept values to the model output. The details are introduced as follows.

***Concept disentanglement.*** To enhance the explainability of the discovered concepts, one important point is to ensure the disentanglement of each concept. To do this, we add a constraint to the distribution of each concept value $p(z_i)$. Following Chen et al. (2018), we use the *Total Correlation (TC)* term to enhance the disentanglement, which forces our model to find statistically independent concepts in the data distribution.

***Explainability of surrogate model.*** To enhance the explainability of the post-hoc explaining process, we further decompose the mapping $h$ (concept values to black-box model outputs) into two steps: (1) Identifying the necessary concepts for each task, (2) predicting the black-box model output with these necessary concepts, as shown in Fig. 2 (b). Since we explain each classifier by its necessary concepts and the concept values, a better explanation will be a smaller number of concepts required by each task. Hence, we can achieve this by ensuring the sparsity of the explanation mask. With identified necessary concepts for each task, we implement the mapping from these concepts to predictions as a type of self-explainable model, soft decision trees, which naturally gives rules for predicting labels with the concept values. We add regularization to the soft decision trees to penalize their complexity for better explainability.

More formally, we decompose $h$ as $h(z) = f_\gamma(m \cdot z)$, $\forall z$, where $m$ is the explanation mask, and $f_\gamma$ is implemented with soft decision trees, parameterized with $\gamma$. We enforce the sparsity of $m$ by penalizing $\|m\|_2$, where $\|\cdot\|_2$ denotes the sum of the square values of the elements. We enforce the explainability by penalizing $C(\gamma)$, where $C(\cdot)$ is a measurement of the complexity of the trees. Then the explainability loss can be written as a weighted sum of the penalty terms for total correlation, sparsity of $m$ and complexity of the decision trees, namely

$$\mathcal{L}_E(x;\phi,m,\gamma) = \underbrace{D_{KL}(q(z)\|\prod_j q(z_j))}_{\text{Disentanglement of } z} + \lambda_3 \underbrace{\|m\|_2}_{\text{Sparsity of } m} + \lambda_4 \underbrace{C(\gamma)}_{\text{Simplicity of } f_\gamma} \qquad (5)$$

where $q(z)$ is the joint approximate posterior, which represents the joint distribution of $z$ over the dataset, and $q(z_i)$ is the marginal approximate posterior, which represents the marginal distribution over the $i-$th concept. $D_{KL}$ denoted the Kullback–Leibler divergence, $C(\cdot)$ is a measurement of the complexity of the trees, $\lambda_3$ and $\lambda_4$ are the weights of the corresponding terms.

**Overall Objective:** All the model components are trained jointly to ensure the discovered concepts are learned with the guidance of both the data and the black-box models. The overall loss function is written as

$$\mathcal{L}(x, f; \phi, \theta, m, \gamma) = \mathcal{L}_I(x; \phi, \theta) + \lambda_1 \mathcal{L}_F(x, f; \phi, m, \gamma) + \lambda_2 \mathcal{L}_E(x; \phi, m, \gamma) \qquad (6)$$

where $\lambda_1$ and $\lambda_2$ are the weight hyper-parameters.

### 4.3 Continuously Improving the Fidelity

The explanation gets more trustworthy when the output of the surrogate model gets closer to the output of the black-box model with the same input. However, it is hard to fully mimic the activities of the black-box models due to limited access to the black-box model's architecture and its training data. Fortunately, our framework supports us to continuously increase fidelity by generating user-customizable and diverse data for training. To achieve this, we devised a strategy to train the model as follows.

To increase the fidelity to mimic a given classifier, we train the model with additional data that is generated by the model itself. The concepts used to general new data can be divided into two groups: the concepts related to this classifier, which can be specified with the user's preference for user-customizability; the concepts

---

**Algorithm 1** Proposed Training Strategy

**Require:** Black-box classifier $f$
**Require:** Trained model parameters $\theta, \phi, m, \gamma$
**Require:** Number of samples of related concepts $n^R$
**Require:** Number of samples of unrelated concepts $n^U$
1: **for** $i = 0$ **to** $n^R$ **do**
2:     Sample $z^R = \{z_j\}_{m_{j,k}=1}$
3:     **for** $j = 0$ **to** $n^U$ **do**
4:         Sample $z^U = \{z_j\}_{m_{j,k}=0}$
5:         $x \leftarrow g_\theta(z^R \oplus z^U)$.
6:         Compute $\mathcal{L}$ with Eq. 6
7:         Update $\theta, \phi, m, \gamma$ with $\mathcal{L}$
8:     **end for**
9: **end for**

---

unrelated to the classifier, which can be perturbated by sampling from the prior distribution for data diversity. Specifically, for the classifier $f_k$ on the $k$-th task, we divide the concepts $z$ into two groups: the set of *related* concepts $z^R = \{z_j\}_{m_{j,k}=1}$, and the set of *unrelated* concepts $z^U = \{z_j\}_{m_{j,k}=0}$. We use the notation $z^R \oplus z^U$ to denote the operation of combining $z^R$ and $z^U$ to a whole set of concepts $z$ while keeping the correct indices. The iterative training process starts by sampling the $z^R$ and perturbating $z^U$ for each $z^R$. Then the data sample $x$ can be generated with the decoder by $x = g_\theta(z^R \oplus z^U)$. With the additional data, the overall objective can be continuously optimized by minimizing $\mathcal{L}(g_\theta(z^R \oplus z^U))$ in Eq. 6.

So the overall objective in the additional training phase can be written as:

$$\min_{\phi, \theta, m, \gamma} \mathbb{E}_{p(z^R)} \mathbb{E}_{p(z^U)} \mathcal{L}(g_\theta(z^R \oplus z^U)) \qquad (7)$$

where $p(z^R)$ and $p(z^R)$ is the distribution of sampling the preferred concept values for generating additional data, $\mathcal{L}$ is the overall objective in Eq. 6.

## 5 Compositional Generalization

Compositional generalization means the ability to recognize or generate novel combinations of observed elementary concepts. One way to achieve compositional generalization is via freezing the trained model weights while training some simple model weights to generalize to new combinations Xu et al. (2022). Our proposed framework, which discovers a set of concepts and identifies the related concepts for each task, can naturally be generalized to explain new black-box classifiers that are unseen in the training phase but the related concepts are found, with all trained model weights frozen, and only train the additional mask and estimators.

Specifically, suppose the model has been well-trained with $k_t$ training tasks $f^{train} = \{f_1, ..., f_t\}$, and $k_c$ concepts are found. We want to use the trained model to explain $p$ unseen test tasks: $f^{test} = \{f_{t+1}, ..., f_{t+p}\}$. The optimization objective is the same with Eq. 4. The difference is that the existing trained weights can be fixed. The only new parameters to train is the added part of the explanation mask $m$, namely $m_{:,t:t+p}$, and the new-added estimators $f_{\gamma}^{test} = \{f_{\gamma_{t+1}}, ..., f_{\gamma_{t+p}}\}$ with trainable parameters $\gamma^{test} = \{\gamma_{t+1}, ..., \gamma_{t+p}\}$. The objective can be written formally as:

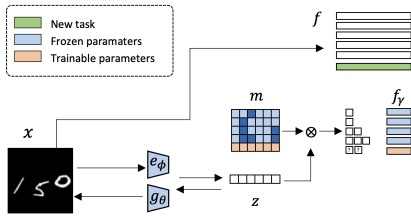

$$\begin{aligned} \underset{m_{:,t+1:t+p},\gamma^{test}}{\text{minimize}} \quad & \mathcal{L}(x, f^{test}) \\ \text{subject to} \quad & \phi, \theta, m_{:,0:t}, \gamma^{train} \quad \text{fixed} \end{aligned} \quad (8)$$

Figure 3: Illustration of the compositional generalization of our proposed framework. Our model can be generalized to new tasks with only training the new parts of the explanation mask $m$ and the estimator for the new task.

where $x$ can be from both the training set or generated with the model as discussed in Sec. 4.3, and $\mathcal{L}$ is the overall objective defined in Eq. 6.

# 6 EXPERIMENTS

In this section, we comprehensively evaluate our proposed method on both concept discovery and post-hoc explanation with qualitative and quantitative results.

## 6.1 EXPERIMENT SETTINGS

**Dataset.** We evaluate our model on the MNIST Deng (2012) dataset and TripleMNIST dataset. For MNIST, following Tran et al. (2022), we select the digits '1, 4, 7, 9' for MNIST dataset. In the TripleMNIST dataset Sun (2019), each image is synthesized by combining three images from the MNIST dataset, with a total of 1000 classes (numbers 000-999). We use 9 classes among them, where each digit is either 0, 1, or 5. We use D1, D2, D3 to represent the first (left-most), second (middle) and third (right-most) digits in the 3-digit number. We developed four black-box classification tasks: $f_1$ for predicting D1, $f_2$ for the parity of the 3-digit number, $f_3$ for whether D2 and D3 are the same, $f_4$ for the value of D1+D2+D3. The black-box tasks as given in Fig. 4 (a). We set $k_c = 6$ for this experiment.

## 6.2 QUALITATIVE EVALUATION

To qualitatively validate the effectiveness of our proposed method, we visualize the discovered concepts in Sec. 6.2.1 and an example of post-hoc explanation on the TripleMNIST dataset to qualitatively

### 6.2.1 DISCOVERED CONCEPTS

To show the semantic meaning of each discovered concept, we conduct interventions on the value of each concept and visualize the generated data. The visualization is shown in Fig. 4.

We visualized the generated data samples by interventions on each concept value in Fig. 4 (b). The inherent semantic meaning of each concept can be attained by observing the variations of the generated data samples. For instance, in the second line (marked with $z_1$) of Fig. 4 (b), the observed variation is the third digit (D3) varies from 0 to 5, then to 1. So the observed semantic meaning of concept $z_1$ is the value of D3. We listed the observed semantic meanings of each discovered concept in Fig. 4 (c). The learned explanation mask $m$ is shown in Fig. 4 (c), representing the related concepts for each task. For instance, for task $f_2$, $z_1$ and $z_5$ are optimized to 1, indicating the concept $z_1$ (D3) and $z_5$ (D2) are related to this task (predicting whether D2=D3).

Results show that, guided by the four classification tasks, our model can discover a set of concepts that have human-understandable semantic meanings while representing the foundational reasoning behind the decisions of each classification task. Our model also successfully identifies the related and unrelated concepts of each task.

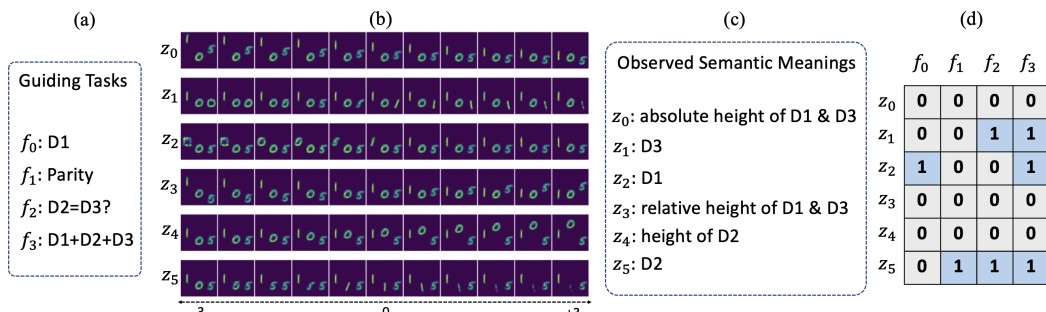

Figure 4: Experiments visualized on the TripleMNIST dataset. (a) The tasks used for guiding the concept discovery. (b) Data samples generated through interventions on individual concepts. Each row alters only the specific concept values indicated, while other concepts remain constant. (c) Semantic interpretations of each discovered concept from variations during concept interventions. (d) Learned explanation mask. For instance, for task $f_2$, $z_1$ and $z_5$ are optimized to 1, indicating the concept $z_1$ (D3) and $z_5$ (D2) are related to this task (predicting whether D2=D3).

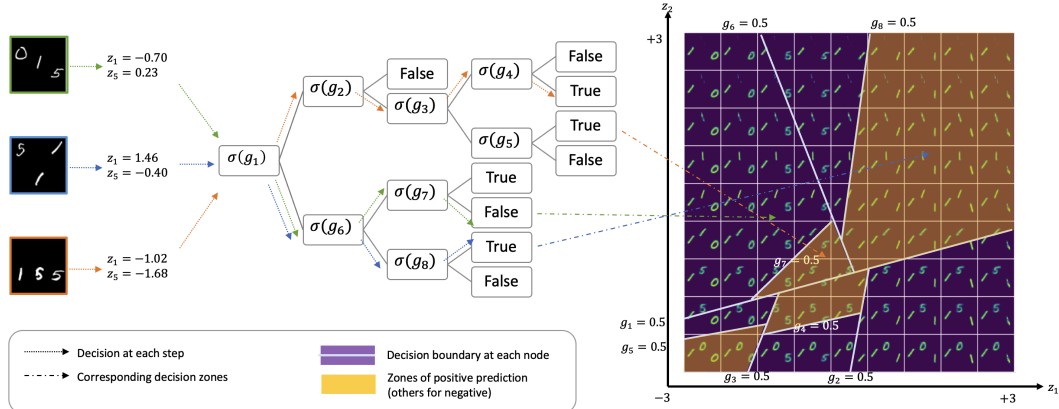

Figure 5: Local explanation for the decision-making progress of three data samples on task $f_2$ on TripleMNIST dataset. (a) The learned decision tree $f_{\gamma_2}$ which maps $z_1$ and $z_5$ to $y_2$. (b) We put the generated images and decision boundaries of the decision tree in the same coordinate. The x-axis represents $z_1$ and the y-axis represents $z_5$, both ranging from -3 to 3. The images are generated with corresponding $z_1$ and $z_5$ according to their position in the coordinates. Each line $g_i = 0.5$ denotes the decision boundary pf node $i$. $\gamma$ denotes the sigmoid function.

### 6.2.2 POST-HOC EXPLANATION

In this subsection, we qualitatively evaluate the post-hoc explanation by taking the explanations on the task $f_2$ for example. The learned concepts and their semantic meanings are the same as in Sec. 6.2.1. The global explanation is shown in Fig. 4 (d). For this case, the learned explanation mask successfully identifies the related concepts of $f_2$ are $z_1$ and $z_5$.

The local explanation of three data samples as examples are shown in Fig. 5 (a). Generally, our proposed method successfully mimics the black-box model's behaviors by first extracting a small number of related concepts, and providing the prediction rules using a simple and transparent model: a decision tree. In the local explanation process, our proposed method first successfully extracts two concepts $z_1$ and $z_5$, that are low-dimensional but enough to reason the decision, compared to the high-dimension original data (82*82). Then the decision is made with a 4-layer decision tree with 8 nodes, and the decision rule of each node is known (for node $i$, the rule is whether $\gamma(g_i) < 0.5$, and $g_i$ is a linear function), yielding a transparent and explainable decision-making process.

In Fig 5 (b), we put the generated images and the decision boundaries of the decision tree in the same coordinates to evaluate the validity of the rules of the decision tree. The results show that the learned linear rules successfully recognize all three zones where $f_2$ is true, corresponding to the three cases that D2 is the same as D3, i.e., D2=D3=0, D2=D3=1, D2=D3=5. Interestingly, two positive

Table 1: Quantitative results of post-hoc explanation on Triple-MNIST dataset.

| Type | Task | #Concept | Depth | #Node | Acc | Acc-S |
|---|---|---|---|---|---|---|
| Test | $f_0$ | 1 | 2 | 2 | 93.61 | 95.96 |
| | $f_1$ | 1 | 2 | 2 | 93.93 | 95.32 |
| | $f_2$ | 2 | 4 | 8 | 88.23 | 94.47 |
| | $f_3$ | 3 | 5 | 27 | 51.61 | 73.20 |
| Generalize | $f_4$ | 1 | 2 | 2 | - | 92.01 |
| | $f_5$ | 3 | 4 | 9 | - | 67.02 |

(yellow) areas in the image's center represent a unique situation where D2=D3=5. This highlights a potential limitation of our approach: the soft decision tree might produce less-than-ideal rules due to inconsistent initialization during its training. We leave this limitation to future work on soft decision trees.

### 6.3 QUANTITATIVE EVALUATION

#### 6.3.1 FIDELITY AND EXPLANABILITY

The evaluations of post-hoc explanations are in Table 1. It presents metrics like recognized concepts (Concept), decision tree depth (Depth), node number (Node), and black-box model accuracy (Acc). Our method mimics black-box models with high fidelity using a transparent model. It translates high-dimensional input to low-dimensional concepts with meaning, then predicts the output. The prediction, via decision trees, is simple with small depth and node numbers. This balance between accuracy and clarity highlights our method's effectiveness in machine learning. The accuracy for $f_3$ is lower due to its 10-class classification nature.

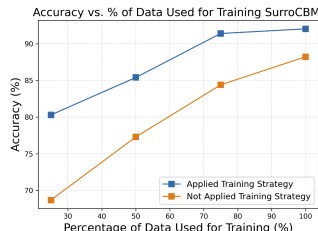

Figure 6: The comparison of data efficacy when using and not using our proposed training strategy.

#### 6.3.2 IMPROVEMENTS FROM ITERATIVE TRAINING

We assessed our training strategy's effectiveness in fidelity improvement, data efficiency, and generalizability. The "Acc-S" column of Table.5 displays the accuracy for each task. Our method notably boosts accuracy, especially for $f_3$ with prior lower performance, proving its efficacy. In Fig.6, we compare data efficacy on $f_1$ to $f_4$. Results indicate greater fidelity improvement with our strategy when training data is limited.

#### 6.3.3 INFORMATION FLOW

To validity that the guidance of task lead to more meaningful discovered concepts, we evaluate the mutual information from each concept to the tasks, calculated by $I(z; y) = \mathbb{E}_{z,y}\left[\log \frac{p(z,y)}{p(z)p(y)}\right]$. We compare the result of our model, with the backbone, beta-VAE(TC), which encourages the disentanglement of each latent factor. Results show that the guidance of classification tasks helps us to find a group of concepts with additionally enforced mutual information to the tasks.

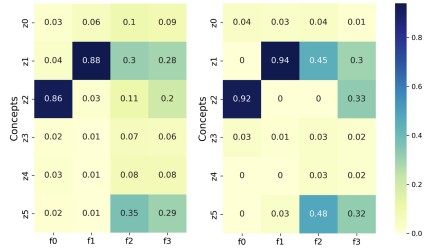

Figure 7: The information flow from each concept to each task. (Left) Beta-VAE(TC). (Right) our model.

## 7 CONCLUSION

In this work, we introduced the Concept Bottleneck Surrogate Models, a novel type of concept-based explainer that can explain black-box classifiers with a set of self-discovered concepts. We propose a training strategy to optimize the model with generated data. The proposed model has the power of compositional generalization. We conducted comprehensive experiments to evaluate the effectiveness of our proposed method.

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

## A  APPENDIX

You may include other additional sections here.

