# OpenReview forum: "SurroCBM: Concept Bottleneck Surrogate Models for Joint Unsupervised Concept Discovery and Post-hoc Explanation"
_ICLR.cc/2024/Conference — ICLR 2024 Conference Withdrawn Submission_

### Official Review · Reviewer_Nbw8 · 2023-10-30

**Soundness:** 3 good
**Presentation:** 3 good
**Contribution:** 1 poor
**Rating:** 3
**Confidence:** 4

**Summary:**

This paper proposes SurroCBM, an approach for post-hoc conceptual explanations of black-box models. The authors first use VAEs to learn concepts and then use a soft decision tree as an explainable surrogate model to shed light on the decision-making process. They suggest additional loss terms to encourage the models to use only a subset of concepts and a data augmentation strategy to enforce alignment of the surrogate model and the black box.

**Strengths:**

**Challenges in conceptual explanations.** The problems of conceptual explanations are well summarized in the introduction. I much agree with the Challenge 1 of current conceptual explanations laid out in the introduction: First, human annotated concepts are hard to obtain, and may not cover all relevant concepts for the prediction. This is a highly relevant (and highly complex) problem.

**Clarity.** Overall, the writing is good to follow apart from some minor issue (see below) and the paper is clearly structured.

**Technical soundness.** I found no severe technical problems with the approach and the notation seems sound in general.

**Weaknesses:**

**Motivation.** The authors unfortunately do not name any relevant real-world tasks where it is essential to predict several characteristics from real-world data. Adding some examples would help to make the setting more relatable and highlight that is relevant. In its current state with the shown MNIST tasks, it seems a bit constructed.

**The evaluation seems too weak.** Unfortunately, only two very small datasets are used in the evaluation that furthermore are both versions of the MNIST dataset. I would have appreciated some experiments on realistic datasets commonly used in the interpretability literature such as the CUB-200 birds dataset. Furthermore, the results still offer room for improvement. In Figure 5b, the parity classifier could potentially be expressed by a single split. However, even with substantially more complexity, the classifier fails to capture many relevant regions. Overall, the results on the synthetic datasets are only mildly convincing, raising questions to whether the method would generalize to practical data. Furthermore, I am missing further ablation studies on the various hyperparameters (e.g., $\lambda_1$ to $\lambda_4$) and how they were chosen in this work.

**No comparison to competing methods.** The empirical evaluation could be further backed up by showing some failure modes of existing approaches, that the presented method does not exhibit. For instance, as now it is only claimed that current surrogate models suffer from a lack of fidelity, there is no evidence to back this up. Fidelity could be measured directly, and the quality of the concepts could be compared at least qualitatively or via some disentanglement metrics for example.

**The overall novelty seems limited.** Coarsely summarizing the paper, the authors propose using a VAE to discover interpretable concepts and then use the soft decision tree as an interpretable classifier on top of those concepts. VAEs are already in common use in conceptual explanations [1, 2]. However, it has recently been argued that generative models cannot be guaranteed to lead to disentangled interpretable factors without additional constraints [3, 5], sparking some criticism on their usage. While I acknowledge that the paper adds some additional loss terms and the data-augmentation technique, I am not convinced by the significance these modifications so far.

**Use of $l_2$ norm to enforce sparsity.** A smaller weakness is that I think the $l_2$-loss may not enforce sparsity. Usually sparsity (corresponding to the “l0”-distance, i.e., the number of non-zero components) is enforced using l1 norms or the l0 pseudo norm or approximations of the l0 pseudo norm such as proposed by Yamada et al [4].


**Minor points**

Related Work: A recent work by Leemann et al. [3] also considers approaches for unsupervised concept discovery and derives conditions when they are identifiable. This could be a potential addition to Section 2.1, as the current paper also touches upon the identifiability problem in Section 3.

Use of citep: Please always use the citep command that puts citations in parentheses (Yeh et al., 2020) when the reference is a grammatical part of the sentence (i.e., the sentence would be read out without the reference).
Intro part of Section 6.2.: there is an unfinished sentence

## Conclusion

A good-to-follow paper on conceptual explanations, which however falls short of the tremendous expectations raised in the introduction in terms of novelty and empirical evaluations. In particular, the empirical evidence has to be backed up by ablation studies and through experiments on additional, realistic datasets. A comparison with other conceptual explanation methods and a real application example could further strengthen the motivation for this particular work.

-------------------------------------------------------
**References**

[1] Thien Q. Tran, Kazuto Fukuchi, Youhei Akimoto, Jun Sakuma: Unsupervised Causal Binary Concepts Discovery with VAE for Black-Box Model Explanation, AAAI Conference on Artificial Intelligence, 2022

[2] O'Shaughnessy, M., Canal, G., Connor, M., Rozell, C., & Davenport, M Generative causal explanations of black-box classifiers. Advances in neural information processing systems (NeurIPS), 2020.

[3] Tobias Leemann, Michael Kirchhof, Yao Rong, Enkelejda Kasneci, and Gjergji Kasneci: When are Post-hoc Conceptual Explanations Identifiable? Conference on Uncertainty in Artificial Intelligence (UAI), 2023.

[4] Yutaro Yamada, Ofir Lindenbaum, Sahand Negahban, and Yuval Kluger. Feature selection using stochastic gates. In Proceedings of the 37th International Conference on Machine Learning (ICML), 2020

[5] Francesco Locatello, Stefan Bauer, Mario Lucic, Gunnar Rätsch, Sylvain Gelly, Bernhard Schölkopf, Olivier Bachem: Challenging common assumptions in the unsupervised learning of disentangled representations.  International Conference on Machine Learning (ICML), 2019

**Questions:**

I have some questions:

1.	Challenge 2 seems a bit disconnected from Challenge 1, which states that in general concepts defined a-priory may not be relevant for another prediction task. What benefits does aligning the dataset concepts with the task have over using a completely different concept set for each predictor as there are no guarantees that the task-relevant concepts may not be entirely different for each task?

2.	What exactly is Acc in Table 5? Is it the accuracy of the black-box model to classify the digits? Or is is the agreement rate between the surrogate model and the original model?

3.	The results in Figure 4d) seem handcrafted. I doubt that a learned matrix would result in exact zero or ones. What were the exact values learned in the matrix and how was the rounding done?

---

### Official Review · Reviewer_SJuA · 2023-10-30

**Soundness:** 2 fair
**Presentation:** 3 good
**Contribution:** 2 fair
**Rating:** 3
**Confidence:** 5

**Summary:**

This paper introduces builds on concept activation vectors (CAVs) and concept bottleneck models (CBMs) to develop concept bottleneck surrogate models (SurroCBM) to explain a black-box model with automatically discovered concepts. The model focuses on post-hoc explanations. Besides, the authors also proposed a training strategy that uses self-generated data to further improve explanations continuously. Preliminary results on Triple-MNIST are provided.

**Strengths:**

Post-hoc explanation of black-box models is an interesting and important topic.

Using concept-based models to explain black-box models is a promising direction beyond saliency-based methods.

The design of the objective function includes three loss terms, each handling identifiability, fidelity, and explainability, which makes sense to me.

**Weaknesses:**

The contribution summary is a bit confusing. I understand the third contribution is distinct from the previous two, but it would be nice if the authors could clarify what is the difference between the first and second points of the contribution summary on Page 2?

One of the key differences between the proposed method and CAVs/CBMs is its concept discovery capability. As the authors mentioned in the related work, there is a rich literature on concept discovery. A lot of them seem to qualify as baselines. However, none of them are included in the evaluation.

This brings me to another one of my major concerns. There are very limited empirical results in the Experiments section. In fact, the authors do not seem to have any baselines, making it difficult to evaluate the contribution of the proposed methodology.

Besides, the authors ran experiments only on MNIST datasets, MNIST and Triple MNIST. These are over-simplified datasets. It would be nice to see how the proposed method works in more complicated datasets, e.g., CUB as used CBMs.

While the identified challenges of discovered concepts in Section 3 makes sense, they are not new; in fact, they are similar to those from “Towards Robust Interpretability with Self-Explaining Neural Networks” (NeurIPS ’18), though the 2018 paper is not on post-hoc explanations. Note that the identifiability challenge is not new either.

Another one of my major concerns is that the proposed method seems incremental, which limited technical merit. For example, in Equation (5), the first two terms are very commonly used in unsupervised learning, e.g., VAE or beta-VAE. The third term and the overall idea of using decision trees is not new either [1, 2, 3]. One could argue that [1, 2, 3] are already automatically discovering concepts using decision trees. In this case, it would actually be reasonable to include them as baselines.

It is also a bit confusing why the paper focuses on a setting with multiple classifiers. Would the method work for typical cases where there is only one classifier? Or is it that the proposed method actually **needs** multiple classifiers to provide sufficient and diverse label information to work?

[1] Extracting decision trees from trained neural networks. KDD 2002

[2] Interpreting CNNs via decision trees. CVPR 2019.

[3] Distilling a neural network into a soft decision tree. 2017.

Minor:

Typo in Page 6: general new data -> generate new data

**Questions:**

The contribution summary is a bit confusing. I understand the third contribution is distinct from the previous two, but it would be nice if the authors could clarify what is the difference between the first and second points of the contribution summary on Page 2?

It is also a bit confusing why the paper focuses on a setting with multiple classifiers. Would the method work for typical cases where there is only one classifier? Or is it that the proposed method actually **needs** multiple classifiers to provide sufficient and diverse label information to work?

---

### Official Review · Reviewer_tyc5 · 2023-10-31

**Soundness:** 1 poor
**Presentation:** 1 poor
**Contribution:** 1 poor
**Rating:** 3
**Confidence:** 4

**Summary:**

This paper introduces Concept Bottleneck Surrogate Models (SurroCBMs), a concept-based explainability framework which learns a globally and locally interpretable surrogate model that can be used to explain a multi-output black-box model of interest. SurroCBMs leverage the power of unsupervised disentanglement learning and feature selection approaches to learn a set of interpretable concepts, and a set of masks indicating which concepts are relevant for a set of tasks of interest, that can be easily adapted to explain models trained on multiple different tasks. These concepts, and their respective task-dependent masks, can be used to explain a classifier's decision through transparent models  (e.g., a soft decision tree). When used in conjunction with experts who may assign semantics to the set of discovered concepts (via, say, interventions), SurroCBMs concept masks enable global interpretability, by highlighting relationships between discovered concepts and labels, and local interpretability, by providing a set of concept-based decision rules that got triggered when making a new prediction. This paper evaluates SurroCBM on an MNIST-based task showcasing that it is able to properly capture known concept-to-label relationships while maintaining a similar or better accuracy than the model they aim to explain.

**Strengths:**

Thank you for submitting this work! I generally found this paper interesting and relevant to the XAI field/community. After a few reads of this work, I believe that the following are its main strengths:

1. **Originality**: Although some of the key components of this paper are not particularly novel or new views/applications/uses of existing methods (see weaknesses below), the overall proposed pipeline can be considered original. Notably, the "compositional generalization" and "continuous fidelity training" parts of this paper are original and worth highlighting.
3. **Quality and Clarity**: As discussed in the weaknesses, the clarity of this paper’s writing and communication could be significantly improved.  Nevertheless, I appreciate this paper’s clear use of diagrams, equations, and algorithm environments to complement some of its text.
4. **Significance**: Because the main SurroCBM pipeline has a lot of moving pieces and is fairly complex, I expect it to have limited significance/impact as it requires significant human interaction (see weaknesses) and potentially a lot of fine-tuning given all its parts. Nevertheless, and connecting this to the originality point made above, the ideas of bootstrapping the surrogate model’s generative process to increase its fidelity and of extracting a set of concepts that can be used in a variety of tasks are both ideas that I can see helping future research develop better frameworks/architectures. Therefore I would vouch for both of these ideas being contributions from this work that may be of benefit for the overall community.

**Weaknesses:**

Taking into consideration the strengths mentioned above, I have the following hesitations regarding this paper in its current state:

1. [Critical] The main proposed SurroCBM pipeline seems to be a conglomeration of several existing methods without much significant innovation in between. The current approach mixes a VAE with a disentanglement regularizer for concept discovery, a feature selection for global explainability, a soft decision tree for local explainability, and a knowledge distillation model for generating a surrogate model to the target model. These are all interesting research components on their own, and putting them together is certainly new. However, the distinction between building an engineering pipeline and providing a new research insight gets a bit blurry when one focuses on connecting a lot of existing methods together. In my opinion, this is never a reason to opt for rejection for a paper if that paper provides significant evaluation, theoretical insights, or interesting takeaways that were extracted from the proposed pipeline. However, as discussed below in more detail, this paper unfortunately is currently lacking significant contributions beyond their proposed pipeline (evaluation is quite limited and no new insights are provided in new datasets, tasks, etc).
2. [Critical] The evaluation is extremely limited, not including **any** other baselines that could be applicable here (e.g., ACE, completeness-aware concept discovery, VAEs and their multiple variants, etc). Furthermore, the model is evaluated on only one simple MNIST-based dataset (e.g., a synthetic dataset only). This leads to a lack of evidence towards the potential merits of the proposed method against existing methods.
3. [Critical] The paper does not solve the problem it set itself to solve in the introduction, that of avoiding human supervision to generate explanations. The need to perform interventions on the discovered latent dimensions of a VAE implies that a human is necessary for this method. This goes against the original intent of the method and weakens the claim that this is less supervision than, say, using a language model to generate concepts (e.g., the manuscript claims that Label-free CBMs are still dependent on human annotations as they are trained using a language model trained on unstructured data such as GPT models).
4. [Critical] Related to the point above, the entire evaluation is mostly qualitative in nature, only including a few quantitative results without any baselines to compare against. This makes it difficult to make judgements that are not affected by confirmation biases.
5. [Critical] The proposed method has four regularization hyperparameters and no ablation studies indicating how the results are dependent on these hyperparameters. In my opinion, the loss is a bit overly complicated and could benefit from simplification as it currently has too many possibly conflicting objectives.
6. [Critical] There are no details on the architectures, models, and hyperparameters used for the reported experiments. Unfortunately, no code or appendix was submitted either. This severely limits the reproducibility of these results and goes against convention in this field.
7. [Critical] The use of surrogate models for explaining a model of interest can be problematic if the surrogate model does not correctly capture the output of the “teacher” model. Evidence presented in the evaluation section of this paper has led me to believe that this is actually happening in this case due to the difference in performance between the surrogate model and the teacher model. If this method is sold as a method to explain a model of interest, then this evidence weakens this claim severely as the surrogate model is being shown as behaving different than the target model we wish to explain.
8. [Major] Against common practice, no error bars are provided in the quantitative evaluation of the proposed method.
9. [Major] The clarity of the manuscript’s writing could significantly benefit from proofreading. As discussed in the question section below, there are multiple typos across the manuscript, many undefined terms used without citations, and several parts that are a bit hard to follow. This is unfortunate given that the ideas in this paper become more challenging to appreciate from the way they are presented.
9. [Major] Some of the figures are too small and low resolution to be able to be clearly seen by the naked eye.
10. [Minor] The related work section could benefit from connecting related work to the proposed method. Currently, this section is enumerating existing papers but not connecting them to the proposed ideas of this paper (see, for example, Section 2.1).
11. [Minor] Forcing discovered concepts to be disentangled may not lead to the explainability goal desired when designing this method. Several concepts, both in concept-annotated datasets and in day-to-day reasoning, are highly entangled and dependent (e.g., “having whiskers” is not fully independent of “having paws” yet they are both important concepts when describing different types of felines and canines).

**Questions:**

Given the critical concerns outlined in the weaknesses above, I am leaning towards rejection of this paper. Nevertheless, the following questions, in no particular order, would help clarify some of the doubts on this work and could help guide the paper to improving on some of the weaknesses detailed above:

1. Enforcing sparsity via an $\ell_2$ loss is an odd choice, given that most feature selection methods opt for an $\ell_1$ penalty instead (e.g., LASSO as a quintessential method). Could you please elaborate on the rationale behind this decision?
2. Regarding my concern about the sensitivity to the many hyperparameters of the objective loss, do you have a sense of how the provided results could change as one modifies these hyperparameters?
3. If the surrogate model performs significantly better than the model itself in the task the model was trained on (as in tasks $f_2$ and $f_3$ in Table 1), how can we be certain that the surrogate model leads to a faithful explanation of the underlying model? Say, if the target model is underperforming for some reason, I would hope the SurroCBM surrogate model to also underperform for that same reason so that when I look at its explanations, I can debug the target model I am interested in in the first place. However, the difference in performances shown in Table 1 seems to suggest that this may not be the case, weakening the main argument of the paper.
4. Fidelity in surrogate models is traditionally measured by how accurately the surrogate model predicts the same output as the model it tries to mimic (regardless of whether the “teacher” model was correct or not in its prediction). Do you have a sense of how your method’s fidelity is when viewed using this traditional metric?

Besides these questions, I found the following typos/minor errors that, if fixed, could improve the quality of the current manuscript:

1. In Section 1, Page 1: “then predict the targets with concepts” should probably be “then predict the targets *from* concepts”
2. In Section 1, Page 1: CLIP should be correctly cited
3. In Section 1, Page 1: “Although some results on concept-based model explanation and concept discovery has been encouraging” should be “Although some results on concept-based model explanation and concept discovery *have* been encouraging”
4. In Section 1, Page 2: it is unclear what “The different requirements to explain the data and classifiers bring a challenge in discovering concepts that meet the post-hoc explanation requirements” means. “Post-hoc explanation requirements” are not well defined up to this point.
5. Section 1, Page 2 (challenge 2 paragraph): citations here and elsewhere seem to be missing the correct format making reading a bit confusing (they should probably be done using \citep{…} in LaTeX)
6. Section 1, Page 2 (challenge 3 paragraph): “Surrogate model-based explanation methods” is undefined up until this point!
7. Section 2.1, Page 2: “Tradition works” should be “Traditional works”
8. nit: Section 2.2, Page 3: ConceptSHAP does not retrain a model as claimed in this section but rather trains a separate model to reconstruct the latent space of a *frozen* model from a set of concept scores it learns to produce.
9. Section 3, Page 3: “then we induce the training objective” should probably be “then we *introduce* the training objective”
10. Section 6.3, Page 9: it seems that default margins in the ICLR format might’ve been accidentally changed, as there is an overlap with Section 6.3’s title and the text above it.
11. Section 6.3.2, Page 9: “Table.5” should probably be “Table 1”
12. Section 6.3.3., Page 9: “To validity” should probably be “To validate”

---

### Official Review · Reviewer_Tpcj · 2023-11-01

**Soundness:** 2 fair
**Presentation:** 2 fair
**Contribution:** 1 poor
**Rating:** 3
**Confidence:** 4

**Summary:**

This paper proposes a framework named SurroCBM, which is designed to discover concepts and then propose explanations (global and local) based on these concepts. SurroCBM contains one concept extractor to discover concepts, and a mapping function to obtain global explanations. A decision tree is trained to provide local explanations. The authors propose an objective function including three parts: identifiability, fidelity, and explainability losses. To further improve the fidelity of explanations, an extra training strategy is proposed. The evaluation of SurroCBM is conducted mainly on TripleMNIST dataset, and the authors analyze the discovered concepts, as well as the fidelity of explanations. The results show that SurroCBM is able to discover meaningful concepts and find global and local explanations for decisions.

**Strengths:**

The paper tackles a challenging research question, namely automatic concept discovery, and proposes a new framework that is trained with several guidance tasks. For instance, four tasks are used in this paper. The designed objective loss is novel in the sense that it incorporates a defined explainablitiy.

**Weaknesses:**

1.	It is not clear that the proposed method can be used to explain other black-box models. It seems that SurroCBM is an interpretable model and can provide concept-based explanations after training. Moreover, there are no details regarding target models in the paper. The author should be clear about his point.

2.	It is not clear how the “black-box classification tasks” are chosen and whether these tasks need extra labels. Since these tasks are very important for concept discovery, the authors should explain more about the impact of these different tasks. If it requires annotations for each task, SurroCBM is then very constrained to concept discovery for complex data input.

3.	Section 4.3 is not well motivated. It seems that the training strategy is proposed to improve accuracy, but not take interpretability into account. Also, how the “related” and “unrelated” concepts are combined needs more clarification. If adding too many “unrelated” concepts into the model, then the interpretability of the whole model should be re-evaluated. In Table 1, there is no regarding information given.

4.	The authors should be straightforward about where the contribution of the SurroCBM stands and compare it to other methods in that area. Experiments on other tasks and comparisons to other methods are needed to support the advantages of SurroCBM. The authors mainly provide experiments on TripleMNIST (no results on MNIST are found). If the authors want to highlight the performance of concept discovery and disentanglement, there are many methods and benchmarks beyond Beta-VAE in Section 6.3.3 that can be compared. If the compositionality is the advantage of SurroCBM, then more analyses on this would be necessary. The audience is also curious about whether SurroCBM can be applied to complex images, such as the bird example given in the teaser figure.

5.	The advantages of local explanations are not strong enough. Although there are only two relevant concepts involved in task f2 in Figure 5. It is still convincing why the local explanations with the decision tree are helpful for users to understand the decision. The authors should motivate and emphasize this point.

6.	There are additional questions that need to be addressed to enhance the clarity of the paper. Please see below.

**Questions:**

1.	Is the decoder in the framework trained to generate new images? The authors have mentioned generated images at many spots but no details about this.

2.	There are six concepts trained (k=6). However, from Figure 6(d), we see that z0, z3 and z4 are not used in any of the four tasks. Any ablation study of only using 4 concepts? Are any performance drops observed in this setting?

3.	On page 8, “In the local explanation process, our proposed method first successfully extracts two concepts z1 and z5”. Does it use the mapping learned from the model? Any results regarding the explanability of other tasks?

4.	Misc: Section 6.3 is overlapped with the paragraph above. The “Acc-S” of Table 1, not Table 5 in Section 6.3.2. The first sentence in Section 6.2 is not finished.